

# Deciphering the genomes of motility-deficient mutants of *Vibrio alginolyticus* 138-2

Kazuma Uesaka[1,2,*], Keita Inaba[1,*], Noriko Nishioka[3], Seiji Kojima[3], Michio Homma[3,4] and Kunio Ihara[1]

[1] Center for Gene Research, Nagoya University, Nagoya, Aichi, Japan
[2] Graduate School of Bioagricultural Sciences, Nagoya University, Nagoya, Aichi, Japan
[3] Division of Biological Science, Graduate School of Science, Nagoya University, Nagoya, Aichi, Japan
[4] Division of Material Science, Graduate School of Science, Nagoya University, Nagoya, Aichi, Japan
* These authors contributed equally to this work.

Corresponding authors
Michio Homma,
g44416a@cc.nagoya-u.ac.jp
Kunio Ihara,
ihara@gene.nagoya-u.ac.jp

## ABSTRACT

The motility of *Vibrio* species plays a pivotal role in their survival and adaptation to diverse environments and is intricately associated with pathogenicity in both humans and aquatic animals. Numerous mutant strains of *Vibrio alginolyticus* have been generated using UV or EMS mutagenesis to probe flagellar motility using molecular genetic approaches. Identifying these mutations promises to yield valuable insights into motility at the protein structural physiology level. In this study, we determined the complete genomic structure of 4 reference specimens of laboratory *V. alginolyticus* strains: a precursor strain, *V. alginolyticus* 138-2, two strains showing defects in the lateral flagellum (VIO5 and YM4), and one strain showing defects in the polar flagellum (YM19). Subsequently, we meticulously ascertained the specific mutation sites within the 18 motility-deficient strains related to the polar flagellum (they fall into three categories: flagellar-deficient, multi-flagellar, and chemotaxis-deficient strains) by whole genome sequencing and mapping to the complete genome of parental strains VIO5 or YM4. The mutant strains had an average of 20.6 (±12.7) mutations, most of which were randomly distributed throughout the genome. However, at least two or more different mutations in six flagellar-related genes were detected in 18 mutants specifically selected as chemotaxis-deficient mutants. Genomic analysis using a large number of mutant strains is a very effective tool to comprehensively identify genes associated with specific phenotypes using forward genetics.

## INTRODUCTION

In biology, genetic approaches assume paramount significance as a method for selecting mutants with specific phenotypic alterations from a vast pool of mutants, concurrently facilitating the identification of the genes responsible for these phenotypes. For instance, the motility apparatus of the marine bacterium *Vibrio alginolyticus* encompasses two

flagellar systems: a single polar flagellum expressed constitutively and multiple lateral flagella induced by shifts in the external environment (*Atsumi et al., 1996*; *Atsumi, McCarter & Imae, 1992*; *McCarter, 2004*). To comprehensively understand the distinct functions of these two flagellar types in motility, we generated mutant strains with deficiencies in either the polar or lateral flagella using EMS and/or UV mutagenesis (*Kawagishi et al., 1995*; *Okunishi, Kawagishi & Homma, 1996*). It is noteworthy that subjecting mutants with polar flagella only to EMS treatment yielded a multitude of motility-deficient strains, furthering our understanding of the molecular structure and function of the polar flagella (*Homma, Nishikino & Kojima, 2022*).

Genes governing flagellar traits in *V. alginolyticus* and its closely related counterpart, *V. parahaemolyticus*, are clustered within several regions of the genome (*Kim & McCarter, 2000*; *Stewart & McCarter, 2003*). Nevertheless, it is pertinent to mention that other genes associated with flagella formation have been reported outside of these demarcated regions (*Brenzinger et al., 2018*; *Yamaichi et al., 2012*). A comprehensive understanding requires a whole-genome analysis of mutants encompassing genes with broader functionality, including those involved in the assembly and positioning of flagellar structures.

In microbial genome analysis, it has been established that short-read sequencing alone permits simultaneous analysis of approximately 300 strains at an approximate cost of USD 3 per strain (*Shapland et al., 2015*). The number of contigs resulting from the assembly may vary depending on the target microbial species; however, in the case of *Vibrio*, assembly of $N_{90}$, which is defined to be the length for which the collection of all contigs of that length or longer contains at least 90% of the sum of the lengths of all contigs, with more than 100 kb, typically yielding fewer than 100 contigs, is achievable (*Castillo et al., 2015*; *Deb, Badhai & Das, 2020*; *LaPorte et al., 2023*; *Meza et al., 2022*). Furthermore, the integration of long-read sequencing enables the relatively straightforward acquisition of complete genomes (*Chen, Erickson & Meng, 2020*; *Miyamoto et al., 2014*; *Wick, Judd & Holt, 2023*). Nevertheless, it should be noted that using long-read technologies, such as PacBio or Oxford NanoPore, entails a substantial cost, often amounting to several hundred dollars per strain.

In this study, we devised a method that exclusively utilizes short-read sequencing for genome assembly. This approach was used to elucidate the complete genome structure of *V. alginolyticus* strain 138-2, a wild-type strain featuring a dual flagellar system, two lateral flagellar-deficient strains (VIO5 and YM4), and a polar flagellar-deficient strain (YM19). Additionally, for the 18 mutant strains derived from the aforementioned flagellar-deficient strains, mapping to the genome of the fully characterized parental strain enabled a comprehensive analysis encompassing all mutation types (single nucleotide variations (SNVs), small insertions/deletions (indels), short tandem repeat number variations (RNVs), and large structural variations (LSVs)) and quantities resulting from EMS or UV mutagenesis. The insights gained from this analysis regarding the types and quantities of mutations induced by EMS or UV treatment provide valuable guidance for future mutant generation. Moreover, this analysis focused on mutants linked to flagellar expression systems, particularly those exhibiting chemotaxis deficiencies (Che⁻ type mutants), leading to the accumulation of mutations in genes integral to regulating flagellar rotation.

Combining random mutagenesis and expression-based selection provides crucial insights into the efficiency of acquiring target gene mutations.

Portions of this text were previously published as part of a preprint (https://www.biorxiv.org/content/10.1101/2023.09.26.559574v2).

## MATERIALS AND METHODS

### Bacterial cultivation and genomic DNA isolation

All *Vibrio alginolyticus* strains used in this study are summarized in Table S1 and the timeline in which the mutants were constructed is summarized in Fig. 1. All mutants have been isolated by focusing on the two types of flagellar systems of *Vibrio*, with strains defective in the Polar flagellum denoted Pof⁻ and those defective in the lateral flagella denoted Laf⁻. Strain VIO5 is a lateral flagella-deficient mutant (Pof⁺, Laf⁻) generated by EMS mutagenesis of VIK4, a spontaneous rifampicin-resistant strain obtained from the wild-type strain 138-2 (Pof⁺, Laf⁺) (*Okunishi, Kawagishi & Homma, 1996*). Strains YM4 and YM5 are lateral flagella-deficient mutants (Pof⁺, Laf⁻) obtained by UV treatment of strain 138-2, whereas strain YM19 is a polar flagellum-deficient mutant (Pof⁻, Laf⁺). This mutant was spontaneously isolated from YM17, a UV-treated YM5-derived strain with impaired polar and lateral flagellar formation (*Kawagishi et al., 1995*). Strain YM51 is a motility-deficient mutant of strain YM5, and strains NMB75, 82, 88, 93, 95, 98, 99, 102, 103, 105, 106, 111, and 116 are motility-deficient mutants of strain YM4 (*Homma et al., 1996*; *Nishioka et al., 1998*). Strains NMB136, NMB155, and KK148 are motility-deficient mutants derived from VIO5 (*Kojima et al., 1999*; *Kusumoto et al., 2006*). All *Vibrio* strains were cultivated in VC medium (0.5% (w/v) tryptone, 0.5% (w/v) yeast extract, 0.4% (w/v) $K_2HPO_4$, 3% (w/v) NaCl, and 0.2% (w/v) glucose). Genomic DNA was isolated in the late logarithmic phase using a genomic DNA isolation kit (Promega, WI, USA).

### Library construction and DNA sequencing

The genomic DNA was quantified by SYBR Green fluorescence method (*Leggate et al., 2006*) and the purity was estimated from the absorption spectra (approximately A260/A280 ratio and A260/A230 ratio) using a NanoVue (GE Healthcare, Chicago, IL, USA). Genomic DNA libraries were constructed by the tagmentation method using the Nextera XT DNA Library prep kit (Illumina, San Diego, CA, USA) or a home-made transposase (*Picelli et al., 2014*), and were sequenced by the MiSeq 600PE v.3 kit or 500PEv.2 kit (Illumina, San Diego, CA, USA). All sequence data were deposited in the DRA Data bank (Accession: DRA012242–DRA012245).

### *De novo* assembly and construction of complete *V. alginolyticus strain* VIO5 genome

*V. alginolyticus* strain VIO5 has been used in many sodium-driven polar flagellar studies (*Li, Kojima & Homma, 2011*). In this study, we first determined the complete genome sequence of strain VIO5. Paired-end sequencing reads (2 × 300-bp) were trimmed by quality value, and adapter sequences were removed using Trimmomatics (*Bolger, Lohse & Usadel, 2014*). Trimmed reads were *de novo* assembled with SPAdes v.3.1 (*Bankevich et al.,*

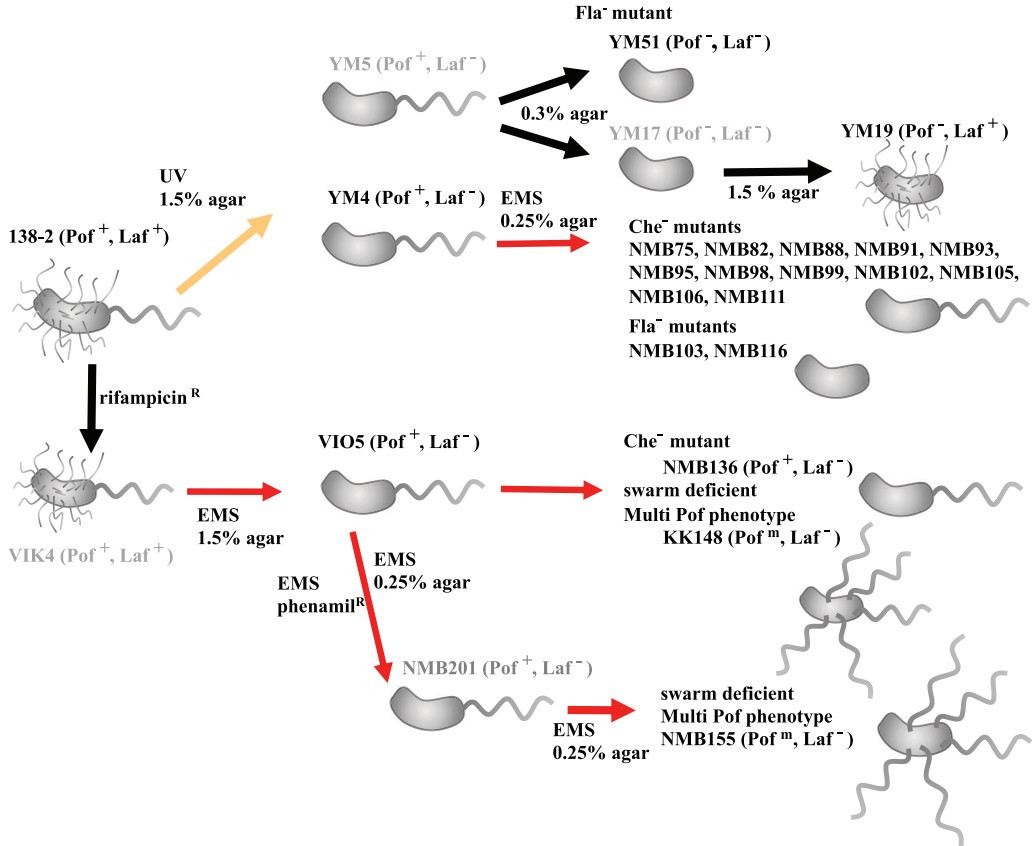

**Figure 1 Procedure for creating motility-deficient mutant strains of *V. alginolyticus* and genealogy of the mutant strains.** *V. alginolyticus* uses two types of flagella depending on the surrounding environment for efficient migration. In low-viscosity aqueous solutions, *V. alginolyticus* swims using a constitutively expressed polar flagellum (Pof), and when viscosity increases, it swarms by inducing the expression of numerous lateral flagella (Laf). Motilities using Pof or Laf can be distinguished on agar plate assay: only bacteria with Laf can swarm on 1.5% agar medium. After selecting Laf⁻ mutant, swim activity with Pof can be evaluated on 0.25% (or 0.3%) soft agar medium by measuring the halo size. Furthermore, the mutants determined to be defective in motility on 0.25% (or 0.3%) soft agar medium were further divided into three categories (Fla⁻, Flaᵐ, and Che⁻) based on flagellar formation or tumbling frequency by microscopic observation. *V. alginolyticus* strain 138-2 (Pof⁺, Laf⁺) was the parent for all the strains in this study. Strain VIO5 (Pof⁺, Laf⁻) was created by EMS mutagenesis of strain VIK4, a spontaneous rifampicin-resistant strain obtained from strain 138-2. Strain NMB136 (Pof⁺, Laf⁻), NMB155 (Pofᵐ, Laf⁻) and strain KK148 (Pofᵐ, Laf⁻) were swim-deficient mutants derived from strain VIO5. Strain YM4 (Pof⁺, Laf⁻) and YM5 (Pof⁺, Laf⁻) were obtained by UV-treatment of strain 138-2 and subsequent selection step on 1.5% agar plate. Strain YM4 was used for the flagella-deficient mutant selection using EMS mutagenesis. Strain YM17 (Pof⁻, Laf⁻) was isolated from a low-concentration (0.3%) agar plate culture of strain YM5 and YM19 (Pof⁻, Laf⁺) was obtained by a subsequent normal-concentration (1.5%) agar plate culture of strain YM17. Strain YM51 (Pof⁻, Laf⁻) was derived from strain YM5. The red lines represent mutagenesis by EMS, the yellow line represents mutagenesis by UV irradiation and the black lines represent spontaneous mutations. All strains used in the genome analysis are shown in black letters, and those not used are shown in gray letters.

*2012*). From approximately 150 assembled contigs, we selected 27 contigs larger than 3 kb whose coverage was close to the most frequent value. Primer 3 (*Rozen & Skaletsky, 2000*) was used to design forward primers specific to both ends of each contig, except for the

repetitive contigs (the primer list can be found in Table S2). The direction and order of the long contigs were predicted by alignment with the genomic sequences of the most closely related *Vibrio* strain (*Vibrio* sp. EX25; accession numbers NC_13456 and NC_13457) using Mauve software (*Darling et al., 2004*). During this operation, the orientation and order of 23 of the 27 contigs were successfully determined. Two contigs formed a circular chromosome (chromosome II) and 21 contigs were linked to form four large assemblies. To determine the linkage order between these four large assemblies and the remaining four contigs that were not aligned in the EX25 genome, PCR experiments were performed using all combinations of primer sets. The amplified PCR fragments were checked for uniformity and size using 0.8% agarose gel electrophoresis and purified using a PCR fragment recovery kit (Promega, Madison, WI, USA). The recovered DNA fragments were sequenced using the MiSeq 500PE v.2 kit (Illumina, San Diego, CA, USA). Short-read sequences from each PCR fragment were assembled using SPAdes v 3.1 (*Bankevich et al., 2012*), and the contig with the largest size and highest coverage was adopted as the PCR fragment. For the three regions in which two large contigs appeared after assembly (The order of contigs are appeared in Table S3), it was inferred that the rRNA operons were arranged in tandem. Therefore, we designed primers in both directions in the center of the connecting contig (approximately 200 bp, contig 175 in Table S4) expected to be between these tandemly arranged rRNA operons, and amplified the two rRNA operons independently by PCR. Both the amplified fragments were independently determined and connected using a central contig. Chromosome I of strain VIO5 was completed by manually combining all contigs and PCR fragment sequences. In regions where discrepancies were found between the contig and PCR fragment sequences, the results of the PCR fragment were used preferentially for linkage because the contig ends may contain polymorphisms due to repeat sequences. Finally, sequencing reads were mapped against the full-length genome sequence using BWA (*Li & Durbin, 2009*) to confirm the absence of assembly errors. The Integrative Genomics Viewer (IGV) (*Thorvaldsdóttir, Robinson & Mesirov, 2013*) was used for map visualization.

## Complete genomes of *V. alginolyticus* 138-2, YM4, and YM19, using *V. alginolyticus* VIO5 as a reference strain

Next, we developed a workflow aimed at completing the genome structure of the target *Vibrio* strains in cases where short-read sequencing technology and very closely related reference strains were available; however, long-read sequencing technology was unavailable because of higher sequencing costs or challenges in high molecular weight DNA extraction. This workflow (for details, see Fig. S1) was applied to assemble the complete genome structures of three strains (*V. alginolyticus* strains 138-2, YM4, and YM19) that are closely related to strain VIO5. Paired-end sequencing reads were trimmed based on their quality using fastp v.0.20.0 (*Chen et al., 2018*) and used as input data (hereafter, WGS reads). WGS reads were assembled *de novo* using SPAdes v.3.13 (*Bankevich et al., 2012*) to produce error-corrected WGS contigs by two cycles of polishing

with Pilon v1. 23 (*Walker et al., 2014*). WGS contigs of less than 1 kbp in length or contigs with abnormal coverage were excluded, and the terminal 127-bp of the remaining WGS contigs were trimmed. These contigs were designated long and normal coverage contigs. For the selection method based on the coverage count, the median of the average read counts of the contigs up to the top five in length was set as value C. Contigs whose coverage was more than twice or less than half of value C were removed as abnormal coverage contigs. Bbmap v.37.62, published by JGI (*SourceForge, 2023*) was used for coverage calculation. LN contigs with a 16-mer frequency greater than or equal to 2 were hard-masked using Primer3_masker (*Kõressaar et al., 2018*). Then, primer3 (*Untergasser et al., 2012*) was used to design outward primers within 1 kb at both ends of each contig. The specificity of the designed primers was checked using FastPCR (*Kalendar et al., 2017*) (the primer list can be found in Table S2). The workflow from short reads to LN contigs and primer design is available on GitHub: script genome_quest (https://GitHub.com/kazumaxneo/genome_quest). Minimap2 (*Li, 2018*) was used to align each LN contig with the complete genome sequence of the *V. alginolyticus* VIO5. PCR was performed using primers designed for each LN contig. Multiplexed PCR products were sequenced using MiSeq and individually assembled, as described in the previous section. LN contigs and Locally Assembled PCR fragment (LA) contigs were connected using CAP3 (*Huang & Madan, 1999.*). Two circular chromosomes were also identified. Finally, WGS reads were mapped to the two assembled chromosomal DNA sequences using minimap2 (*Li, 2018*), to correct as many errors as possible using the following variation detection tools: breseq (*Deatherage & Barrick, 2014*), GATK HaplotypeCaller v 3.8 (*DePristo et al., 2011*), minimap2 paftool (*Li, 2018*), and SV Quest (v1.0) (https://GitHub.com/kazumaxneo/SV-Quest).

The complete genome sequences of the four strains have been published in the database under the accession numbers AP022859–AP022866 (DDBJ).

## Variation analyses in the mutant strains

Genomic libraries were constructed and sequenced by MiSeq using paired-end sequencing (2 × 300 bp) for 17 strains generated by EMS mutagenesis and one strain, YM51, derived from YM5. Using the genome of *V. alginolyticus* 138-2 as a reference, all variants (SNVs, indels, RNVs, and LSVs) present in the three strains were extracted using Equation (*Deatherage & Barrick, 2014*) and other tools, as described in the previous section.

## Core gene phylogenetic tree inference

Each mutant genome was created using the gdtools APPLY command from the output of the Breseq variant calling against the 138-2 genome sequence (*Deatherage & Barrick, 2014*). Core genome alignment of the 138-2 sequence and the derived sequence of the mutant strain was performed using Parsnp (https://GitHub.com/marbl/parsnp). Unreliable alignment blocks were excluded based on the Parsnp criteria. A phylogenetic tree was manually constructed based on the number of variations.

## RESULTS

### Genome structures of *V. alginolyticus* strain 138-2 and derived mutant strains

We determined the complete genome structure of *V. alginolyticus* strain 138-2 and three mutant strains: VIO5, YM4, and YM19. These strains have been widely used for the functional analysis of polar flagella of the genus *Vibrio* (*Kawagishi et al., 1995*). Similar to the genomes reported for other *Vibrio* spp., the genomic DNA consists of two circular chromosomes and does not harbor any plasmids (*Okada et al., 2005*). The genome size was 5,185,395 bp for strains 138-2 and VIO5 and 5,185,324 bp for strains YM4 and YM19. DFAST annotation predicted 4,601 protein-coding sequences (CDSs) for strain VIO5, 4,602 for 138-2, and 4,603 for YM4 and YM19. Thirty-seven rRNA genes (twelve 16S rRNA, twelve 23S rRNA, and thirteen 5S rRNA) and one hundred and sixteen tRNA genes were assigned to all four strains (detailed genomic information is provided in Table S5).

### Variation sites in *V. alginolyticus* strains VIO5, YM4, and YM19

*V. alginolyticus* strain VIO5 is a lateral flagellar-deficient mutant that arises from EMS mutagenesis of VIK4 (rifampicin-resistant), which results from a spontaneous mutation in the parent strain 138-2 (Fig. 1). The VIO5 strain has four SNVs compared to its parent strain 138-2, three on chromosome 1 and one on chromosome 2 (Table 1). The rifampicin-resistant phenotype of VIO5 can be attributed to a mutation in chromosome I (position 3,206,619) of the VIO5 genome (Table 1). This mutation leads to a Q513L amino acid substitution in the RNA polymerase beta subunit, which has been reported as a causal SNP of rifampicin resistance in *E. coli* (*Campbell et al., 2001*). Two other SNVs in chromosome 1 are on the *hslO* gene and Vag1382_04350 gene, which are predicted to encode heat shock 33 kDa chaperonin and glutamate synthase, respectively. Neither of these two genes is thought to be involved in the lateral flagellar deficient phenotype of VIO5. A mutation on chromosome II (position 1,534,464) introduced a stop codon at codon 64 in *motY2* (TGG→TGA). Given that this is the sole mutation observed in chromosome II of VIO5, and considering that all lateral flagellar genes are present on chromosome II, the lateral flagellar deficiency was attributed to a mutation in the *motY2* gene.

*V. alginolyticus* strain YM4, a lateral flagellar-deficient mutant, and strain YM19, a polar flagellar-deficient mutant, were generated through UV mutagenesis of strain 138-2 (Fig. 1). Thirteen mutations were common between YM4 and YM19 (Table 1), suggesting that these mutations accumulated during the early stages of UV mutagenesis. Since YM4 and YM19 exhibit different phenotypes for the two types of flagella, a mutation on chromosome II (position 255,362), exclusive to YM4, resulting in a serine substitution at Gly53 in the lateral flagellar P-ring protein FlgI, is the likely cause of the YM4 lateral flagellar-deficient phenotype. Similarly, a mutation on chromosome I (positions 2,299,407), found only in YM19, introduced a stop codon at codon 295 (CAG→TAG) in *flhA*, which encodes the polar flagellar export apparatus protein and presumably resulted in a truncated FlhA and the polar flagellar-deficient phenotype of YM19.

Uesaka et al. (2024), PeerJ, DOI 10.7717/peerj.17126

**Table 1 Variation sites in the three mutant strains compared to the parental strain 138-2.**

| Position | 138-2 | VIO5 | YM4 | YM19 | Mutation | Annotation | Description |
|---|---|---|---|---|---|---|---|
| Chr. I | | | | | | | |
| 135,143 | C | T | C | C | D100N (GAT→AAT) | hslO | 33 kDa chaperonin |
| 165,966 | T | T | A | A | D328V (GAT→GTT) | Vag1382_01380 | Bifunctional GTP diphosphokinase guanosine-3′,5′-bis(diphosphate) 3′-diphosphatase |
| 213,963 | G | G | C | G | Q180E (CAA→GAA) | Vag1382_01870 | 3-deoxy-D-manno-octulosonic acid transferase |
| 415,183 | | | Δ11 bp | Δ11 bp | Y397-FS | tolC | Outer membrane channel protein |
| 485,114 | A | C | A | A | I137S (ATC→AGC) | Vag1382_04350 | Glutamate synthase |
| 582,644 | A | A | C | C | M256L (ATG→CTG) | phoR | PAS domain-containing sensor histidine kinase |
| 592,588 | (TCAT)$_2$ | (TCAT)$_2$ | (TCAT)$_1$ | (TCAT)$_1$ | I240-FS | pstB1 | Phosphate import ATP-binding protein PstB 1 |
| 1,174,956 | G | G | T | T | A16E (GCA→GAA) | Vag1382_10490 | DNA-binding protein |
| 1,787,856 | (ACCAA)$_5$ | (ACCAA)$_5$ | (ACCAA)$_6$ | (ACCAA)$_6$ | T451-FS | Vag1382_16050 | Hypothetical protein |
| 1,885,712 | (TGTTTT)$_2$ | (TGTTTT)$_2$ | (TGTTTT)$_1$ | (TGTTTT)$_1$ | Intergenic (−206/−323) | ihfA Vag1382_16940 | Integration host factor subunit alpha Membrane protein |
| 2,065,000 | | | Δ57 bp | Δ57 bp | Δ19aa | Vag1382_18450 | Membrane protein |
| 2,289,413 | (GCTCTG)$_9$ | (GCTCTG)$_9$ | (GCTCTG)$_{10}$ | (GCTCTG)$_{10}$ | PE repeat 9→10 | Vag1382_20480 | Chemotaxis protein CheW |
| 2,299,407 | G | G | G | A | Q295* (CAG→TAG) | flhA | Flagellar biosynthesis protein FlhA |
| 2,585,439 | T | T | G | G | Q73H (CAA→CAC) | Vag1382_23340 | Hypothetical protein |
| 2,701,438 | | | +TA | +TA | I198FS | sfsA | Sugar fermentation stimulation protein |
| 3,204,405 | C | C | T | T | S1251N (AGC→AAC) | rpoB | DNA-directed RNA polymerase subunit beta |
| 3,206,619 | T | A | T | T | Q513L (CAG→CTG) | rpoB | DNA-directed RNA polymerase subunit beta |
| Chr.II | | | | | | | |
| 255,362 | G | G | A | G | G53S (GGC→AGC) | flgI2 | Flagellar P-ring protein 2 FlgI |
| 513,122 | (AAAAAT)$_3$ | (AAAAAT)$_3$ | (AAAAAT)$_2$ | (AAAAAT)$_2$ | Intergenic (+160/−50) | Vag1382_35310 Vag1382_35320 | MFS transporter Methyl-accepting chemotaxis protein |
| 1,534,464 | C | T | C | C | W64* (TGG→TGA) | Vag1382_44100 | Flagellar protein MotY2 |

**Note:**
From the left column, genomic position, base status of each of 138-2, VIO5, YM4, and YM19, mutation (amino change in case of coding region), annotation (conventional name or strain name), and description (protein function inferred by the annotation).

The VIO5 strain generated by EMS mutagenesis had only four SNVs with no insertion or deletion mutations, but the YM4 and YM19 strains created by UV mutagenesis had two deletion mutations and five repeat number variation mutations, in addition to seven and six SNVs for YM4 and YM19, respectively (Table 1).

## Variation sites in other motility-deficient mutant strains

*Vibrio* strains NMB136, NMB155, and KK148 were generated from strain VIO5, whereas strains NMB75, NMB82, NMB88, NMB93, NMB95, NMB98, NMB99, NMB102, NMB103, NMB105, NMB106, NMB111, and NMB116 were generated from strain YM4. All 16 strains were generated through EMS mutagenesis and screened as mutants that could not form a swimming ring on a soft agar plate; motility was observed under dark-field microscopy (*Homma et al., 1996*). In Table S6 summarizes all detected mutation sites in the genome of each strain compared with the 138-2 strain genome. The mutations were categorized into three types: single nucleotide variations (SNVs), short insertions/ deletions (indels), and short tandem repeat number variations (RNVs). Considering these variations, the number of SNVs, indels, and RNVs detected in each strain was used to create a pedigree for the strain (Fig. 2). Because NMB136, NMB155, and KK148 were generated from the VIO5 strain at different times, these three strains carried completely independent mutations. Conversely, the 14 NMB strains generated from YM4 almost simultaneously carried nine common mutations (five SNVs, two indels, and two RNVs) in addition to various unique mutations ranging from two to 54. Considering the individual strains analyzed, the mutations they carried were counted independently, resulting in an average of 20.6 ± 12.7 mutations (mean ± standard deviation).

## Putatively responsible variations for motility-deficient mutant strains

The motility-deficient mutants analyzed in this study can be classified into three types: those with no or incomplete flagella (Fla⁻ type), those with chemotaxis problems (Che⁻ type), and those with an increased number of polar flagella (Pof^m type). Another type, the Mot⁻ type, which has abnormalities in the rotation apparatus, was absent in the analyzed mutants. These three types of motility-deficient mutants had 10–30 variation sites, but all had mutations in a known flagella-related gene (Figs. 3A and 3B). The Fla⁻ type mutants, NMB103 and NMB116, had mutations in the *flgL* gene, leading to the observation of only the hook structure without visible flagellar filaments, which explains their flagella-deficient phenotype. Che⁻ type mutants included NMB75, NMB82, NMB88, NMB91, NMB93, NMB95, NMB98, NMB99, NMB102, NMB105, NMB106, NMB111, and NMB136. NMB82 and NMB105 harbored mutations in the *cheA* gene, NMB91 and NMB98 harbored mutations in the *zomB* gene, NMB93 and NMB136 harbored mutations in the *cheY* gene, and NMB88, NMB95, NMB99, NMB102, and NMB106 harbored mutations in the *fliM* gene. These genes are part of the gene cluster responsible for the chemotactic response in *Vibrio*, especially the change in flagellar rotation (Fig. 4). Pof^m type mutants included the KK148 and NMB155 strains. The KK148 strain harbored a mutation in the *flhG* gene, whereas the NMB155 strain harbored a mutation in the *fliM* gene.

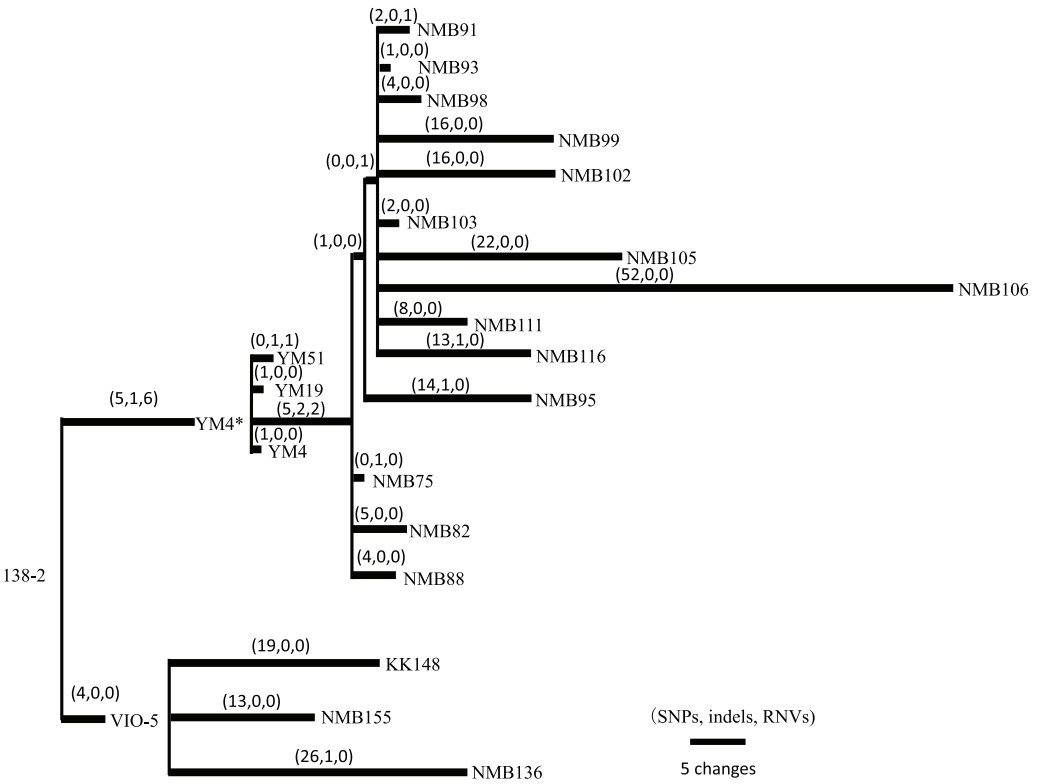

**Figure 2  Pedigree tree based on the SNVs, indels, and RNVs among *V. alginolyticus* mutant strains.**
Illustration of the phylogeny of the mutant strains, assuming the mutations (SNVs, indels, and RNVs) to
be equidistant. The three numbers in parentheses drawn above the branches represent SNVs, indels, and
RNVs. YM4* is the true parental strain of the 14 NMB mutants, and the YM4 strain whose genome was
analyzed has a single SNP which has occurred after NMB mutats creation experiment.

## DISCUSSION

### Responsible genes for lateral flagellar-deficient mutants

In each of the three lateral flagellar-deficient strains (VIO5, YM4, and YM51), the reasons
for the flagellar-deficient phenotype were different. In the VIO5 strain, the lateral flagellar
deficiency was attributed to a mutation in the *motY2* gene. Intriguingly, a single mutation
in a structural gene can result in the complete loss of flagellar gene expression. The *motY2*
gene was one of the earliest members to be expressed in the lateral flagellar expression
hierarchy (*Stewart & McCarter, 2003*) and was positioned at the head of the operon
(Fig. 3B). Therefore, a mutation leading to a premature stop codon in the *motY2* might
significantly impact the translation of downstream *lafK* genes, a $\sigma^{54}$-dependent regulator
required for the expression of class 2 genes of lateral flagella (*Stewart & McCarter, 2003*).
In the strain YM4, a mutation in the *flgI* gene may be strongly related to lateral flagellar
deficiency. This gene product FlgI is the major component protein that forms the P ring
located in the periplasmic region, and when P ring formation is incomplete, protein
transport that constitutes hook and flagellin fibers is impaired and normal flagellin
formation is inhibited. In the P-ring protein FlgI of *E. coli*, a point mutation such as G21C

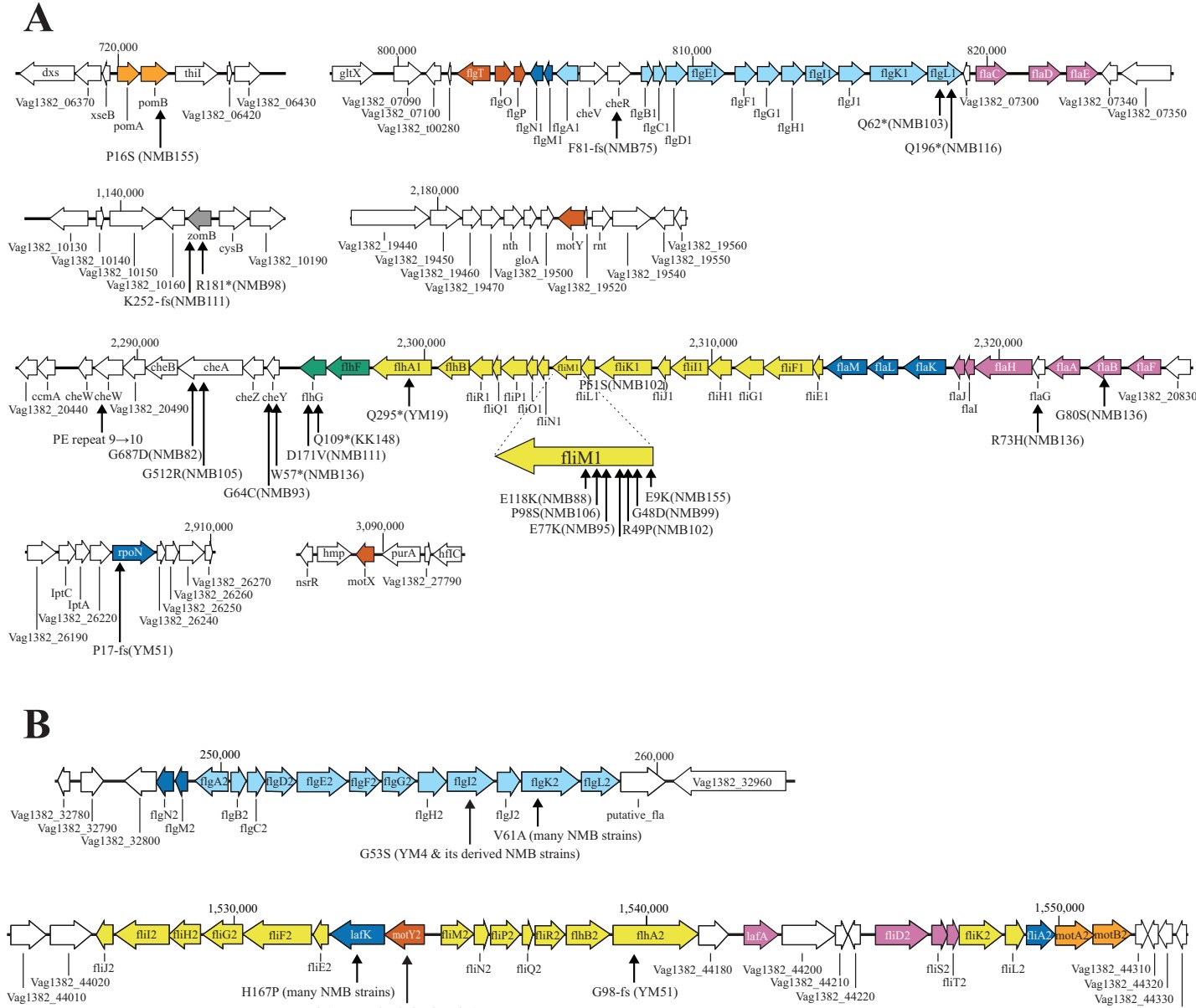

**Figure 3 Known flagella-related genes in chromosome I (A) and chromosome II (B) and the detected variations in mutant strains.** Chromosome I contains a cluster of genes related to the polar flagellum and chemotaxis signal transduction (A), and chromosome II contains a cluster of genes related to the lateral flagella (B). Chromosome I consists of seven regions: two regions consisting of large clusters and five regions consisting of one or two genes; chemo-signal transduction genes are found in one of the two large clusters. In addition to this, chemoreceptors are present scattered throughout the genome. Chromosome II consists of two large clusters in two regions. Known genes are listed by their customary names, and the paralogs of the lateral hair genes on chromosome II are distinguished by the four letters, followed by the number 2. The arrow extended below the gene indicates the location of the mutation that occurred in that gene. A mutation is an amino acid change found in the protein, and the name of the strain in which the mutation was found is provided in parentheses. The genes involved in flagellar formation are divided into seven categories, and each group is indicated by a different color coding; blue (expression regulation), reddish purple (filament, cap, chaperone), sky blue (hook, L-ring, P-ring), brown (T-ring and H-ring), yellow (MS-ring, C-ring, and T3SS), vermilion (stator/motor) and bluish green (flagellar number regulation).

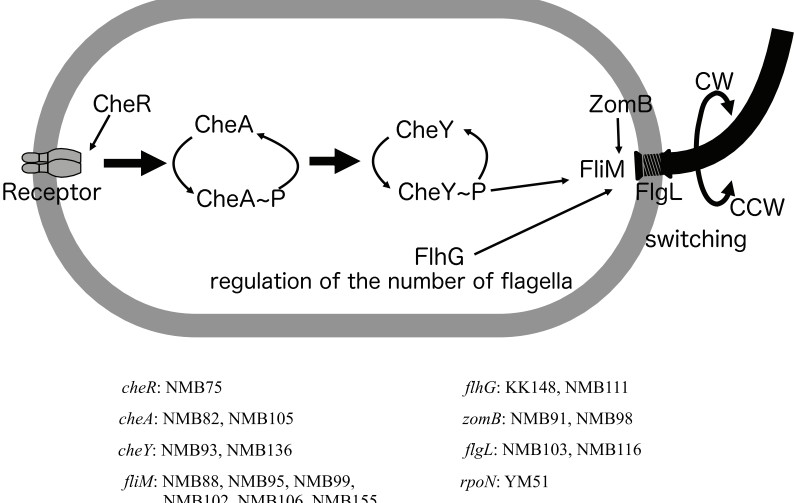

cheR: NMB75

cheA: NMB82, NMB105

cheY: NMB93, NMB136

fliM: NMB88, NMB95, NMB99,
    NMB102, NMB106, NMB155

flhG: KK148, NMB111

zomB: NMB91, NMB98

flgL: NMB103, NMB116

rpoN: YM51

**Figure 4 Schematic diagram of the chemoreceptor signaling pathways and regulation of flagellar rotation in a mutant strain with only a polar flagellum.** Several components to control the flagellar rotation from chemoreceptors to flagella were schematically depicted. Of the 18 mutants analyzed, two had mutations in the flagellar structural gene (*flgL*), 2 in the gene controlling the number of flagella (*flhG*), and 13 in genes involved in the transmission of information from chemoreceptors to control the direction of flagellar rotation (*cheR, cheA, cheY, fliM, zomB*). The remaining one strain had a mutation in a sigma factor (*rpoN*) involved in the transcription of a group of flagellar-related genes. Of these eight genes, all but the *cheR* and *rpoN* genes were found to have different mutations in two or more strains. The names of the genes and the mutant strains with them are listed in the bottom.

causes a complete loss of motility (*Hizukuri et al., 2008*), so it is not surprising that a mutation in FlgI(G51S) of strain YM4 suppresses expression of lateral flagellar. It has also been shown that in *Salmonella* when the anti-sigma factor FlgM is not expelled by the protein transport hook-basal body apparatus, the sigma factor FliA does not function (*Hughes et al., 1993*; *Kutsukake, 1994*) and transcription of the class 3 genes involved in flagellar formation does not occur (*Aldridge et al., 2006*). It remains to be elucidated to what extent lateral flagellar formation occurs in the strain YM4. In the strain YM51, a frame-shift mutation in the *flhA2* gene (Fig. 3B) may be strongly related to lateral flagellar deficiency. YM51 was originally isolated from YM5 (Fig. 1) and exhibits a Fla⁻ type phenotype. In the *flhA* gene region, seven consecutive G bases were found to have increased by one, causing truncation of the FlhA protein to approximately 100 amino acids. In *Paenibacillus glucanolyticus*, it was reported that swarming was suppressed by reversible hotspots that reduced the number of eight consecutive A bases to seven and that the strain could easily revert to a swarm-competent state (*Hefetz et al., 2023*). Thus, there appear to be two types of lateral flagella-deficient *Vibrio* mutants: a strong phenotype with almost irreversible mutations, such as VIO5 and YM4, and a leaky phenotype that is relatively easy to revert, such as YM51. A causal mutation in YM5's lateral flagellar-deficient phenotype may be the same as in strain YM51.

## Responsible genes for polar flagellar-deficient mutants

Polar flagellar-deficient mutants can be divided into three categories based on microscopic observations of movement: Che⁻ mutants, in which the direction of flagellar rotation is dysregulated; Pof^m mutants, in which the number of polar flagella is increased; and Fla⁻ mutants, in which have no flagella at all.

Among the Che⁻ type mutants, the *fliM* gene, identified as the causal gene for many mutations, appears to be involved in flagellar rotation control and the regulation of flagellar numbers (*Homma et al., 2022*) (Figs. 3A and 4). This suggests versatile roles of the *fliM* gene in governing flagellar expression. The NMB75 shows to smoothly swim with reduced response to phenol (*Homma et al., 1996*). The NMB75 strain harbored a mutation in the *cheR* gene, resulting in a leaky phenotype due to partial signal transmission by CheA, partially affected by the deficiency of CheR activity in methylating chemoreceptors. Strain NMB88, NMB95, NMB99, NMB106 and NMB136 swim smoothly without much tumbling by locking the direction of flagellar rotation to CCW. Strain NMB102, on the other hand, has its flagella locked in the CW direction of rotation and swims constantly backward. NMB136 has a defective mutation of CheY (nonsense mutation of W57), which may be the reason for the observed CCW-locked flagellar motion. NMB88, NMB95, NMB99, and NMB106 all have mutations in FliM, possibly weakening its interaction with phosphorylated CheY, which is required for tumbling. On the other hand, in strain NMB102, R49P, one of the two mutations in FliM, has been shown to be important for the CW-locked phenotype, in which structural changes in FliM itself result in a CheY-independent rotational motion fixed in the CW direction (*Takekawa et al., 2021b*). NMB111 showed weakly reduced swimming ability due to mutations in the *flhG* gene, which regulates flagellar number, and flagella were rarely observed with the FlhG(D171A) mutation (*Ono et al., 2015*). The reduced swimming ability of NMB111 may be due to the reduced flagellar number caused by the FlhG(D171N) mutation. Thus, the NMB111 strain may be included in the Fla⁻ type mutants.

In two Pof^m type mutants, the KK148 strain harbored a mutation (Q109*) in the *flhG* gene and a defective *flhG* gene product has been shown to form multiple polar flagella (*Kusumoto et al., 2006*). Whereas the NMB155 strain harbored a mutation (E9K) in the *fliM* gene, and it has been shown that the FliM(E9K) mutation changes flagellar numbers (*Homma et al., 2022*). Based on the evidence, the fact that the *fliM* gene, which has been implicated in chemotaxis, also plays a significant role in regulating flagellar number in the Che⁻ type mutants is highly intriguing.

The YM51 strain, similar to NMB103 and NMB116, showed only a hook structure without visible flagellar filaments, indicating that the assembly of the polar flagellar filament was impaired (*Nishioka et al., 1998*). However, no mutations were found in flagellar structure genes, including the *flgL* gene, but a mutation was detected in the *rpoN* gene, which has been reported to play an important role in polar flagella formation (*Kawagishi et al., 1997*). Although the YM14 strain used for cloning the *rpoN* gene was not included in the current genome analysis, it is highly plausible that YM51 has a mutation similar to that of YM14.

## Comparison of two flagellar systems

Since the two entire flagellar systems, the polar and lateral flagellar systems, are homologous to each other, it is very interesting to examine the similarity of each flagellar gene and whether there are other paralog genes with similar functions in the genome. Therefore, for the 104 flagellar-related genes that appeared in Fig. 3, we examined the paralog genes in the genome of strain 138-2 by amino acid homology and compared the corresponding genes in the polar and lateral flagella, which are summarized in Table 2. Many paralog genes were detected in the genome for the genes constituting the Che protein group of the signal transduction system (CheA, CheY, CheV, CheW, *etc.*), however, there were no functionary paralogous genes for flagellar-motility system that showed full-length homology, but limited to a few regions such as histidine kinase domain or response regulator domain(A list of paralogs of flagellar-related genes found in the genome is provided in Table S7). This is inferred from the fact that only the cheY gene mutation located in the flagellar gene cluster region (Fig. 3) resulted in a motility-deficient phenotype. The flagellar-related genes are divided into two groups: genes encoding proteins that form the flagellar structure and genes encoding proteins that regulate the expression of flagellar genes. In both cases, the paralogs of the two flagellar systems were the most closely related genes in the genome, and the only multiplicated genes were flagellin genes encoding the polar flagella (A list of amino acid identities among the seven flagellar proteins is provided in Table S8). In the two flagellar systems, a certain degree of amino acid identity was observed between the proteins constructing the hook, L-ring, P-ring, rod, MS-ring, C-ring, and T3SS (type three secretion system) (Table 2). However, proteins related to the regulation of flagellar gene expression, proteins constituting the T-ring and H-ring, and proteins constituting the Stator-Motor showed very low amino acid identity, and in many cases, no homologous protein was found in the lateral flagellum (Table 2). If the two flagellar motor systems are somewhat similar, it is possible that homologous proteins, especially those constituting the T-ring and H-ring, are located elsewhere on the chromosome. Genome analysis of a large number of motility deficient mutants using a strain deficient in the polar flagellum (YM19) will most likely reveal genes involved in lateral flagellar formation.

## The potential of powerful forward genetics as a comprehensive analysis of systems

Through genomic analysis of 18 mutant strains selected by a combination of EMS mutagenesis and screening for motility-deficient phenotypes, it was found that these strains contained 3 to 75 gene mutations in addition to the one or two gene mutations presumed to be strongly related to the phenotype (motility). Gene manipulation to selectively disrupt (or introduce mutations in) specific genes is necessary to determine the gene responsible for the phenotype, and in *Vibrio alginolyticus*, several genes have been identified as flagellar-related genes by mutagenesis (*Homma et al., 2022*; *Kitaoka et al., 2013*; *Takekawa et al., 2021a*, *2021b*). In this study, six genes involved in the regulatory system of flagellar rotation were enriched (*i.e.*, several different types of mutations were concentrated in the same gene) in the analysis of only 18 strains (Fig. 4), suggesting the

**Table 2 Comparison of paralogous proteins of two flagellar systems.**

| Category[#1] | Polar flagella | | | % Identity[#2] | Lateral flagella | |
|---|---|---|---|---|---|---|
| | Gene name | Function | aa length | | Gene name | aa length |
| Expression Regulation | fliA1 | RNApol sigma28 | 244 | 30.4% 74/224 | fliA2 | 242 |
| | flaM | Two-component response regulator | 469 | 42.6% 231/467 | lafK | 443 |
| | flaK | Sigma54 dependent regulator | 488 | [#3] | [#3] | – |
| | flaL | Two-component sensor kinase | 343 | [#3] | [#3] | – |
| | flgM1 | Anti sigma28 factor | 104 | ND | flgM2 | 93 |
| | flgN1 | Molecular chaperone for FlgK, L | 141 | 50% 10/20 | flgN2 | 145 |
| | rpoN | RNApol sigma54 | 489 | [#4] | [#4] | – |
| Fillament, Cap, Chaperone | flaC | Flagellin | 384 | [#5] | lafA | 281 |
| | flaD | Flagellin | 377 | [#5] | | |
| | flaE | Flagellin | 374 | [#5] | | |
| | flaA | Flagellin | 376 | [#5] | | |
| | flaB | Flagellin | 377 | [#5] | | |
| | flaF | Flagellin | 377 | [#5] | | |
| | flaG | Flagellin accessory | 144 | [#4] | [#4] | – |
| | flaH | HAP2 filament cap | 663 | 25% 145/460 | fliD2 | 445 |
| | flaJ | Flagellin chaperone | 136 | 30% 36/119 | fliS2 | 128 |
| | flaI | Unknown | 101 | ND | fliT2 | 106 |
| Hook, Rod, L-ring, P-ring | flgA1 | P-ring chaperone | 248 | 27.7% 77/217 | flgA2 | 265 |
| | flgB1 | Proximal rod | 131 | 38.5% 61/130 | flgB2 | 120 |
| | flgC1 | Proximal rod | 137 | 41.5% 64/142 | flgC2 | 144 |
| | flgD1 | Hook assembly | 236 | 27.2% 53/147 | flgD2 | 227 |
| | flgE1 | Hook | 437 | 34.3% 202/440 | flgE2 | 398 |
| | flgF1 | Proximal rod | 249 | 41.9% 109/246 | flgF2 | 243 |
| | flgG1 | Distal rod | 262 | 54.6% 144/262 | flgG2 | 261 |
| | flgH1 | L ring | 259 | 36.9% 88/198 | flgH2 | 223 |
| | flgI1 | P ring | 363 | 49.2% 180/362 | flgI2 | 373 |
| | flgJ1 | Peptidoglycan hydrolase | 307 | 34.7% 41/95 | flgJ2 | 182 |
| | flgK1 | HAP1 hook-filament junction | 646 | 28.5% 104/365 | flgK2 | 457 |
| | flgL1 | HAP3 hook-filament junction | 397 | 33.0% 93/282 | flgL2 | 299 |
| T-ring, H-ring | motX | T-ring | 211 | [#4] | [#4] | – |
| | motY | T-ring | 293 | 25.6% 66/247 | motY2 | 339 |
| | flgO | H-ring | 377 | [#4] | [#4] | – |
| | flgP | H-ring | 212 | [#4] | [#4] | – |
| | flgT | H-ring | 143 | [#4] | [#4] | – |
| MS-ring, C-ring, T3SS | flhA1 | Fla export | 710 | 50.4% 361/697 | flhA2 | 696 |
| | flhB1 | Fla export | 376 | 38.9% 157/368 | flhB2 | 375 |
| | fliE1 | Hook basal body MS ring-rod junction | 103 | 35.2% 26/71 | fliE2 | 118 |
| | fliF1 | MS-ring | 580 | 27.1% 200/535 | fliF2 | 569 |
| | fliG1 | Rotor/switch component | 351 | 30.5% 99/321 | fliG2 | 337 |
| | fliH1 | Fla export; negative regulator of FliI | 266 | 29.2% 68/195 | fliH2 | 251 |

(Continued)

| Category[#1] | Polar flagella | | | | | Lateral flagella | |
| --- | --- | --- | --- | --- | --- | --- | --- |
| | Gene name | Function | aa length | % Identity[#2] | | Gene name | aa length |
| | fliI1 | Fla export; ATPase | 439 | 53.5% 238/437 | | fliI2 | 448 |
| | fliJ1 | Fla export | 147 | ND | | fliJ2 | 146 |
| | fliK1 | Hook-length control | 627 | 26.6% 34/109 | | fliK2 | 357 |
| | fliL1 | Unknown | 167 | 27.2% 33/114 | | fliL2 | 166 |
| | fliM1 | C ring switch | 348 | 22.1% 89/204 | | fliM2 | 272 |
| | fliN1 | C ring switch | 136 | 50.7% 38/75 | | fliN2 | 123 |
| | fliO | Fla export | 119 | [#3] | [#3] | – | |
| | fliP1 | Fla export | 289 | 56.7% 150/240 | | fliP2 | 252 |
| | fliQ1 | Fla export | 89 | 54.1% 46/85 | | fliQ2 | 89 |
| | fliR1 | Fla export | 260 | 34.3% 81/230 | | fliR2 | 258 |
| Stator/Motor | pomA | Stator/force generator | 253 | ND | | motA2 | 285 |
| | pomB | Stator/force generator | 315 | 33.1% 55/157 | | motB2 | 330 |
| Flagellar number regulator | flhG | Flagella number positive regulator | 295 | [#4] | [#4] | – | |
| | flhF | Flagella number negative regulator | 495 | [#4] | [#4] | – | |

**Notes:**
[#1] These seven categories are consistent with the color coding in Fig. 3.
[#2] Identity (%) and matched number of amino acids over total number of amino acids compared were depicted and ND stands for Not Detected.
[#3] Many paralogous proteins were detected other than the lateral flagellar system (See Table S7).
[#4] No paralogous proteins were detected in the genome.
[#5] All seven flagellins are paralogous to each other (See Table S8).

possibility of comprehensively extracting genes in the entire system by increasing the number of mutant strains analyzed. Since most of the 18 strains were selected for the Che⁻ phenotype (*Homma et al., 1996*), it is possible that this selection pressure narrowed down the number of genes to a few genes out of approximately 50 genes involved in a polar flagellum formation. Thus, genome analyses of a large number of completely random, independent motility-deficient mutants could extract additional genes involved in the flagellar system (if the genes are not essential for growth). This means that even for microorganisms that cannot be genetically manipulated, a sufficient amount of mutant strain analysis, using a well-designed selection pressure, could efficiently detect the various phenotypic components that build a living system.

## CONCLUSIONS

In this study, in the genome analysis of mutants created by the selection for reduced chemotaxis ability, it was possible to show that mutations were concentrated in a few genes among the about 50 genes that make up the flagellar system, even in the genome analysis of only a dozen or so mutants. In the past, genetic analysis of mutants itself was time-consuming and labor-intensive, and screening had to be devised to limit the number of analyzed strains, but with the current efficient and low-cost genome analysis method, it is possible to analyze as few as 1,000 strains. Analysis of a large number of mutant strains is

expected to greatly advance our understanding of the phenotype of microbial species, especially those that are difficult to genetically manipulate.

## ACKNOWLEDGEMENTS

This study utilized mutants developed in a series of studies on the structural and functional analysis of *Vibrio* polar flagella driven by Na⁺ electrochemical potential differences. This research endeavor was initially initiated by Prof. Imae and was subsequently carried forward by two authors, Homma and Kojima. We extend our gratitude to Kawagishi, Okunishi, Maekawa, Kusumoto, and Yorimitsu for constructing the mutants. We would like to thank Editage for English language editing.

### Funding

This work was supported by JSPS KAKENHI, Grant Number 19K06627, to Kunio Ihara and Grant Number 20H03220, to Michio Homma. The funders had no role in study design, data collection and analysis, decision to publish, or preparation of the manuscript.

### Grant Disclosures

The following grant information was disclosed by the authors:
JSPS KAKENHI: 19K06627, 20H03220.

### Competing Interests

The authors declare that they have no competing interests.

### Author Contributions

- Kazuma Uesaka performed the experiments, analyzed the data, prepared figures and/or tables, and approved the final draft.
- Keita Inaba performed the experiments, analyzed the data, prepared figures and/or tables, and approved the final draft.
- Noriko Nishioka performed the experiments, prepared figures and/or tables, and approved the final draft.
- Seiji Kojima conceived and designed the experiments, authored or reviewed drafts of the article, and approved the final draft.
- Michio Homma conceived and designed the experiments, authored or reviewed drafts of the article, and approved the final draft.
- Kunio Ihara conceived and designed the experiments, performed the experiments, analyzed the data, prepared figures and/or tables, authored or reviewed drafts of the article, and approved the final draft.

### DNA Deposition

The following information was supplied regarding the deposition of DNA sequences:
The complete genome sequences of the four strains (Vibrio alginolyticus strains 138-2, VIO5, YM4, YM19) described here are available at DDBJ: AP022859 to AP022866.

## Data Availability

All sequence read data used in this study are also available *via* DRA Data bank (Accession: DRA012242–DRA012245, DRA016647, DRA016649).

The workflow in this study is available on GitHub: script genome_quest and SV-Quest genome_quest

URL: https://github.com/kazumaxneo/genome_quest

DOI: 10.5281/zenodo.10633276

SV-Quest

URL: https://github.com/kazumaxneo/SV-Quest

DOI: 10.5281/zenodo.10633286

All detected mutation sites are available in the Table S6.

## Supplemental Information

Supplemental information for this article can be found online at http://dx.doi.org/10.7717/peerj.17126#supplemental-information.

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
