# Peer review of "Deciphering the genomes of motility-deficient mutants of Vibrio alginolyticus 138-2"

_PeerJ, doi:10.7717/peerj.17126_

## Round 0.1 · original submission · Major Revisions

Please revise the manuscript by following the reviewers' comments. Point-by-point responses are needed when re-submitting your manuscript for re-consideration.

Reviewer 1 ·

Basic reporting

For the most part, the manuscript is well written and easy to follow. There are places though where the language should be changed for clarification. Such places include: In line 61, the authors should describe what is meant by the phrase “assembly of N90 with more than 100 kb”. The authors should define the Pof and Laf phenotypes the first time they are introduced in line 87. On line 208, “causing the 64th tryptophan residue of the MotY2 protein” could be misconstrued as the 64th tryptophan residue in the protein. The same goes for “53rd glycine to serine” on line 220. The description of the lineage of strain YM19 in lines 214-216 is confusing. In line 261, the statement, “Each strain exhibits various degrees of fixed rotation in its expression system” is unclear. Figure 3 is difficult to read as the font size is small. Magnifying the image does not aid in reading the print as it is very pixelated.

Complete genome sequences for four Vibrio alginolyticus strains are reported here and accession numbers for the sequences are provided.

Experimental design

The research falls within the Aims and Scope of the journal. The research question is clearly defined as identifying the genetic lesions that are responsible for motility and chemotaxis defects in V. alginolyticus mutants that were generated in the study. V. alginolyticus has two separate flagellar systems (i.e., polar and lateral flagella) that require dozens of genes for their assembly and function, and there are significant gaps in our knowledge of these genes and how the two systems are interconnected. The authors identify mutations in known flagellar and chemotaxis genes in strains that are deficient in motility or chemotaxis, but this information does not provide any new knowledge of motility and chemotaxis in V. alginolyticus. Moreover, there is no attempt to confirm that the mutations that identified from genome sequencing were responsible for the observed phenotypes of the mutants. There are a couple of interesting mutations (e.g., flgI mutation in YM4, and the fliM mutation in NMB155, and the flhG mutation in NMB111) that are reported here. However, given that the strains have multiple mutations, it is important to introduce the mutations into a naive background to confirm that they are responsible for the observed phenotypes of the original mutants. The experimental design is described adequately for the genome sequencing. There is not description of how chemotaxis was assessed, and the authors seem to have examined the flagellation phenotypes of some of the mutants, but there is no description on how this was done either.

Validity of the findings

As indicated above, the authors do not how chemotaxis assays were done or the basis for ascribing a chemotaxis deficiency to the mutants. It is unclear how the authors distinguish between chemotaxis deficiency and reduced motor function. The conclusions for the genetic basis for the motility and chemotaxis defects in many of the strains is likely well supported given that these strains have mutations in well characterized flagellar genes. For other strains, the conclusions regarding the genetic lesions that are responsible for the observed phenotypes of the strains is premature given that the lack of verification.

Additional comments

no comment

Reviewer 2 ·

Basic reporting

no comment

Experimental design

The manuscript presents genetic analyses of a number of Vibrio alginolyticus strains that have been used a lot in the field because of their flagellar/motility phenotypes, but whose genotypes had been unknown. The study is very valuable because it enables an improved interpretation of existing and future research results on these strains.

I am not competent to judge the methodology.

Validity of the findings

I verified that I am able to access data under the given identifier, but I am not competent to assess it.

The general flow of the arguments strike me as sound.

Additional comments

1. The abstract would benefit from a more specific wording. “exhaustive analysis of genomic structures of Vibrios strains” is a bit general. Specific information that should be added:
a. Species information, not only Genus
b. How many strains and which? (WT 138-2 and xx descendants?)
c. What exactly was done? Whole genome sequencing for all strains?
d. Does “motility-deficient” mean fully non-motile, or does it include defects in either polar or lateral flagella?
2. Ll. 71 – 76: It is unclear to me what the difference in approach was for the 17 mutants compared to the previously mentioned strains. Presumably whole genomes were determined for all? Please specify.
3. The authors attempt to attribute the strains’ known phenotypes to specific mutations.
a. While the proposed explanations strike me as highly plausible and very likely to be true, I missed a specific explanation of the route of deduction. Specifically, how many other genetic differences to a closely related strain with a different phenotype were found, and on what basis are those deemed less likely to explain the phenotypic feature? E.g. ll 218-221.
b. Given that the proposed explanations (though highly plausible) are not backed up by experiments (e.g. phenotypic confirmation of clean deletions of those specific genes in WT background), I would also suggest to reword accordingly in the discussion. E.g. not “a mutation in the motY2 gene is responsible for the lateral flagellar deficiency” (ll 267-8), but, for example, “the lateral flagellar deficiency was attributed to a mutation in the motY2 gene”. And similar for the other statements in the discussion.
4. In some places, I struggled to determine the general structure of the approach. The text contains references to supplementary material that is not well explained. In particular, are the various tables complete lists of all items of a certain type, or a selection based on specific criteria, and if the latter, what are the criteria? The problem is compounded by the fact that I could not find captions (or sometimes even titles) for some of the Supplementary Tables. For the reader it would be useful if the reference to the SI in the main text already indicated what type of data is in there, and if the title and caption then spelled out the details.
5. Ll 343-345: Why would other homologous genes elsewhere on the chromosome not have been found in the present analysis, given that the whole genome sequence is available?
6. In my version of the manuscript, the text in Fig. 1, in particular the superscripts in the phenotypic description, are not or barely legible. That should be improved.
7. If I am not mistaken, intermediate ancestors such as YM5 and VIK4 were not analyzed.
a. While I don’t think it is necessary to do so, it would be useful to explain briefly why these were not analyzed genetically, given that the potential genetic basis of their phenotypes is discussed in the paper.
b. It might be useful to the reader to graphically distinguish strains that were analyzed and those that were not in Fig. 1, e.g. by using grey text for the latter.
8. Might there be a way to design Figs 1 and 2 more similar to each other in layout to ease a comparison between genealogy and genetics?
9. The discussion would benefit from paragraphs to separate analyses of different strains.
10. I would encourage the authors to include S. Fig. 2 (helpful schematic of the chemotaxis and motility apparatus with its constituent proteins) in the main article, if possible, to support their arguments on the functional impact of mutations.
11. Is V. alginolyticus particularly difficult to manipulate genetically? If yes, that would be worth mentioning in the introduction or discussion.
12. Ll. 261 : Missing references for statement?
13. Ll 268-9: Missing reference supporting the statement? Besides, MotY2 is not a structural protein, neither is LafK. What is meant here?
14. Ll 330-334: I could not follow the argument. What feature of Fig. 3 is referred to here?

---

## Round 0.2 · Minor Revisions

Please revise the manuscript by following the reviewer's suggestions. I would need a point-by-point response letter when re-submitting your manuscript.

Reviewer 1 ·

Basic reporting

The authors have adequately addressed my previous comments regarding basic reporting, with the exception of the wording for the mutations. In the authors' rebuttal, they indicate that the 64th tryptophan of MotY2 "literally means the 64th tryptophan". If this is true, that means that tryptophan residue in MotY2 is preceded by 63 tryptophan residues. I suggest changing the sentences in question to the following or something similar:
LIne 224 of revised manuscript with marked changes - "A mutation on chromosome II (position 1,534,464) introduced a stop codon at codon 64 in motY2 (TGG -> TGA)."
Line 237 - change "leading to a change in the 53rd glycine to serine" to "resulting in a serine substitution at Gly53"
Line 240 - change "mutated the 295th glutamine of the polar flagellar expoort apparatus protein FlhA to a termination codon" to "introduced a stop codon at codon 295 (CAG ->TAG) in flhA, which encodes the polar flagellar export apparatus protein and presumably resulted in a truncated FlhA that accounted for the polar flagellar-deficient phenotype of YM19".

Experimental design

The authors have adequately addressed my previous comments regarding sufficient description of the methods.

Validity of the findings

The authors have adequately addressed my previous comments regarding how deficiencies in chemotaxis and motor function were distinguished, and about overstating their conclusions.

Additional comments

no comment

---

## Round 0.3 · accepted · Accept

The authors have addressed the minor-revision comments. The manuscript is acceptable for publication now.